# When I am sixty-four... evaluating language markers of well-being in healthy aging narratives

Tabea Meier[1,2,3,4]*, Matthias R. Mehl[5], Mike Martin[1,2,3,6,7], Andrea B. Horn[1,2,3,6]*

1 Department of Psychology, University of Zurich, Zurich, Switzerland, 2 University Research Priority Program (URPP) "Dynamics of Healthy Aging", University of Zurich, Zurich, Switzerland, 3 Healthy Longevity Center, University of Zurich, Zurich, Switzerland, 4 School of Education and Social Policy, Northwestern University, Evanston, Illinois, United States of America, 5 Department of Psychology, University of Arizona, Tucson, Arizona, United States of America, 6 Center for Gerontology, University of Zurich, Zurich, Switzerland, 7 Faculty of Health and Behavioral Sciences, School of Psychology, The University of Queensland, Brisbane, Qld, Australia

* t.meier@psychologie.uzh.ch (TM); a.horn@psychologie.uzh.ch (ABH)

**Data Availability Statement:** The data and analysis scripts used in the present article are available on the Open Science Framework: https://osf.io/s3qr6/.

**Funding:** Funding by the Jacobs Foundation (https://jacobsfoundation.org/en/; awarded to TM)

## Abstract

Natural language use is a promising candidate for the development of innovative measures of well-being to complement self-report measures. The type of words individuals use can reveal important psychological processes that underlie well-being across the lifespan. In this preregistered, cross-sectional study, we propose a conceptual model of language markers of well-being and use written narratives about healthy aging ($N = 701$) and computerized text analysis (LIWC) to empirically validate the model. As hypothesized, we identified a model with three groups of language markers (reflecting affective, evaluative, and social processes). Initial validation with established self-report scales ($N = 30$ subscales) showed that these language markers reliably predict core components of well-being and underlying processes. Our results support the concurrent validity of the conceptual language model and allude to the added benefits of language-based measures, which are thought to reflect less conscious processes of well-being. Future research is needed to continue validating language markers of well-being across the lifespan in a theoretically informed and contextualized way, which will lay the foundation for inferring people's well-being from their natural language use.

## Introduction

In recent years, there has been increasing interest in inferring individuals' health and well-being from their language. The type of words people use reflect a variety of psychological processes, including their thinking or coping styles [1–4], negative emotional tendencies and depression [5]. Natural language use offers promising opportunities for tracking well-being alongside traditionally used self-report questionnaires. From a developmental perspective, well-being can be seen as the outcome of the life-long process of healthy aging [6].

and the Swiss National Science Foundation SNF ((https://www.snf.ch/en/; fellowship P2ZHP1_199409 awarded to TM), as well as by the "Gerontopsychology and Gerontology" unit, Department of Psychology, University of Zurich, Switzerland (https://www.psychology.uzh.ch/en/areas/dev/geronto.html) helped to conduct this research. The preparation of this manuscript was further supported by a grant from the National Institutes of Health (https://www.nih.gov; U19AG065169 awarded to MRM). The views, opinions, and findings contained in this document are those of the authors and should not be construed as position, policy, or decision of the aforementioned agencies, unless so designated by other documents. The funders had no role in study design, data collection and analysis, decision to publish, or preparation of the manuscript.

**Competing interests:** The authors have declared that no competing interests exist.

Importantly, healthy aging represents a shared human experience that takes place across the whole lifespan and does not only start once people are old [6]. The World Health Organization (WHO) has put the validation of new measures of well-being on top of their priority list, as part of their efforts to monitor and promote healthy aging [6]. In the present study, we used computerized language analysis of personal narratives as an innovative, automated way to assess well-being and its underlying psychological processes. With the rise of the internet and social media, language data are more available than ever and have vast potential to reveal insights into psychological phenomena such as well-being [5,7]. Expressive writing studies, for example, have shown that individuals' changes in health and well-being over time are reflected in subtle changes in their word use throughout writing sessions [8,9]. More recent studies suggested that language patterns may serve as early indicators of mental health problems and underlying risk factors (e.g., increased self-focus) [5,10,11].

While existing language studies provide a promising fundament, most studies have linked separate language markers (e.g., self-references) to separate aspects of well-being (e.g., life satisfaction). What is currently missing is a conceptual integration of language markers that reflect well-being and underlying processes, while acknowledging the multi-dimensional structure of well-being. The present study aims to provide a more comprehensive examination of language and well-being, informed by a lifespan perspective of healthy aging and empirical studies on language use. Individuals of all ages have thoughts and feelings about aging [12,13], and these personal perspectives seem to be broadly comparable across age groups [12]. Aging views held at younger ages may be especially important as they can turn into self-fulfilling prophecies and affect individuals' actual aging when they become older [14]. The present study used a writing paradigm in which an age-diverse sample expressed their personal views of "aging well" to empirically test associations between theorized language markers and well-established well-being dimensions. Specifically, the study aims to (a) introduce a conceptual model of language markers and (b) link language markers to established self-report scales of well-being dimensions. Personal writings about aging represent a suitable testing ground for language markers of well-being. Particularly, reflecting upon one's desired or expected aging trajectory should activate relevant processes that underlie well-being across the lifespan and manifest as subtle differences in language use [7,11,15].

## Affective, evaluative, and social components of well-being: Theoretical foundations

An individual's word use can reveal what they are paying attention to and how they are feeling [7,16]. Importantly, it reveals much more than the mere content of the message. The field of psychological language analysis has distinguished between so-called *content words* (e.g., sun, love), which indicate *what* a person is disclosing, and *function words* (typically small words such as pronouns or articles), which indicate *how* they are saying it (i.e., language style) [7]. The same message can be delivered in different styles, and it is often the non-conscious, stylistic differences that are closely tied to relevant processes of well-being. It has, for example, been shown that depression can accurately be detected from subtle stylistic cues in self-descriptions [17]. Our conceptual working model of language indicators is informed by the vast body of empirical work [7] that has linked word use with various aspects of well-being and integrated it into current well-being concepts. Specifically, prior work has identified several language markers of affective [5], cognitive-evaluative [1,11,15], and social processes [18,19], suggesting them as valid candidate variables for relevant aspects of well-being. Before formulating a conceptual language model, it is important to emphasize that language provides a different type of access to individuals' inner worlds than conventional self-report measures. Specifically,

whereas questionnaires explicitly ask individuals to report on their well-being—tapping into consciously accessible aspects—, language provides a much more indirect way—tapping into the less conscious processes that contribute to individuals' well-being [16,20]. Thus, even if language use predicts self-reported well-being (and vice versa), it is possible that the very structure of these measurement types differs from one another. In other words: Language markers of well-being and underlying processes may not necessarily show the same structure that fundamental well-being theories—which have mainly relied on self-report data—have proposed. To integrate our language model into the well-being literature, we briefly review some fundamental well-being theories, focusing on affective, evaluative, and social aspects that are likely to manifest in language use [7,20].

Well-being, broadly defined as "optimal psychological functioning and experience" [21], is generally agreed upon to be a multi-dimensional construct that encompasses affective, evaluative and social aspects [22,23], despite vast differences in specific conceptualizations. The well-being literature has placed great emphasis on different subcomponents. Some researchers have, for example, distinguished between affective (i.e., the positive and negative emotions people tend to experience) and subjective well-being (i.e., how positively people evaluate their life and functioning; [22,24]). Others have highlighted so-called eudaimonic aspects, focusing on what makes life truly meaningful and enables people to thrive (e.g., purpose in life, self-realization) [23], which is particularly important for healthy aging [25]. While often seen as separate components by proponents from different traditions [21], subjective [22] and eudaimonic well-being [23] converge in that they both represent evaluative aspects of well-being—cognitive evaluations of various areas of life—rather than the mere experience of emotions (i.e., affective well-being) [21]. This notion is supported by a large, cross-cultural investigation [26] showing that subjective and eudaimonic well-being can be summed up in one overarching component (i.e., evaluative well-being), which may include life satisfaction (or related constructs such as quality of life), and evaluations about relevant domains of functioning (e.g., subjective physical health, purpose in life). In spite of the long-standing research tradition and broad consensus on the presented dimensions of well-being, debates about how to conclusively frame the multi-faceted nature of well-being are ongoing [27].

Beyond affective and evaluative components of well-being, social relationships are clearly important for well-being. The formation of enduring close relationships is a core human need [28] and loneliness a key risk factor for well-being [29] and longevity [30,31]. For example, a large longitudinal study found a higher mortality risk and poorer cognitive and physical functioning for older adults with smaller and less diverse social networks [32]. Within the well-being literature, many conceptualizations have emphasized the importance of satisfying social relationships [23,33], with some seeing social functioning as its own component that is distinct from affective or evaluative well-being [23,29]. Especially as people age, social relationships are thought to become even more important [34,35]. Relations with others (e.g., spouse, family, friends) can critically shape aging trajectories [36] and have gained special emphasis in lifespan perspectives on well-being [23,33,35].

To summarize, integrating the literature on natural language use (which we review in detail below) and well-being across the lifespan, we frame well-being as a multi-dimensional construct that includes affective, evaluative and social features of human functioning. We do not claim these components to be entirely separate from one another, as empirical work often shows correlations between these factors [21]. Rather, we see them as interrelated, yet distinct, components that build a useful conceptual framework to examine our hypothesized language markers.

### Language markers of well-being: Empirical evidence

In the following, we briefly review previous evidence that linked word use to *affective*, *evaluative* and *social* processes of well-being, which will lay the ground for our language-based model of well-being.

### Affective well-being

Emotion words (e.g., happy, sad, depressed) reflect an individuals' focus on positive or negative emotions [37] and have, as one might intuitively assume, indeed been linked to affective well-being [17,38]. Beyond emotion words in a narrow sense, we expect words referring to rewarding processes (e.g., opportunity, succeed) to be linked to affective well-being, since goal pursuit and fulfillment play an important role for well-being and healthy aging [6]. Furthermore, self-referential language (i.e., I-talk), i.e., using a high rate of first person singular pronouns, has reliably been linked to depression, negative affectivity, and vulnerable narcissism and thus lower affective well-being [5,39,40]. Additionally, words that amplify emotional experiences, so-called quantifiers (e.g., few, many, very), have been linked to depressive symptoms and lower affective well-being [41]. In summary, we expect that individuals with greater affective well-being will use more references to positive emotional processes, fewer self-references, and fewer quantifiers in their writings.

### Evaluative well-being

Language markers intuitively related to evaluative well-being include those that indicate cognitive elaboration. Substantial cognitive elaboration is necessary to make sense of complex topics —such as reflecting upon one's well-being and healthy aging—and can be seen as an underlying process of evaluative well-being [3,15]. Language styles characterized by complexity and cognitive elaboration are thought to reflect adaptive thought processing modes, contrasting with abstract and decontextualized thought processing modes [42], which are considered maladaptive and often involved in mental health problems [43]. Expressive writing studies, for example, showed that participants with the greatest health benefits over the course of writing were those who increased their use of cognitive processes words (e.g., to realize, understand) [15]. Similarly, patients who exhibited greater cognitive elaboration in initial trauma narratives showed attenuated post-traumatic symptoms later on [11]. Beyond cognitive elaboration, a psychologically distant mode of thought processing is considered adaptive self-reflection that may prevent from emotional overinvolvement [2,44]. On the linguistic level, cognitive elaboration is indicated by cognitive processes words (e.g., to realize, understand, because) and interrogative words (e.g., why?, when?) [15,45]. Additionally, a more analytical thinking style (indicated by the use of articles or prepositions rather than personal pronouns or conjunctions) can indicate both cognitive elaboration and psychological distance and can be captured by a coherent language dimension of analytical thinking [1,46,47].

To conclude, we expect language markers indicative of cognitive elaboration and psychological distance (e.g., cognitive processes words, interrogatives, analytical thinking) to reflect processes that underlie greater evaluative well-being. This seems particularly relevant in the context of healthy aging narratives, as the perception and anticipation of own aging requires the identification and integration of potential risks, gains, and sources of meaning, thus requiring deeper cognitive elaboration.

### Social well-being

In addition to affective and cognitive-evaluative processing styles, the words people use can reveal their perceptions of social integration [48,49]. Past research shows that people with

greater social well-being make more references to social processes and other people. For instance, in a study of student photo essays, individuals who were more socially integrated used more social words (e.g., friends, family) [50], and, in a daily diary study, participants with greater daily well-being used more social words and pronouns referring to other people (e.g., he, she, they) [19]. Additionally, words referring to affiliation (e.g., team, together) [51] and communal orientation (we-pronouns) [18] have been associated with social well-being and related processes [18,51]. In the present study, we expect that word choices referring to other people (e.g., social processes, family friends, she/he, they-pronouns) and social integration (e.g., affiliation words, we-pronouns) underlie a social dimension of well-being [29,35].

## The present study

As new language-based approaches [52,53] and empirical evidence to link word use with facets of well-being continue to accumulate, there is great potential for language analysis research to predict and better understand individuals' well-being. What is missing to move this research forward is a more systematic evaluation of language markers of well-being, embedding them into a theoretical framework. In the present study, we propose a conceptual working model of language markers (see Fig 1). The goal is to identify a theoretically informed internal structure of language variables and link them to established measures of well-being (self-report scales), which represents a first attempt at validating language indicators of well-being in the context of healthy aging.

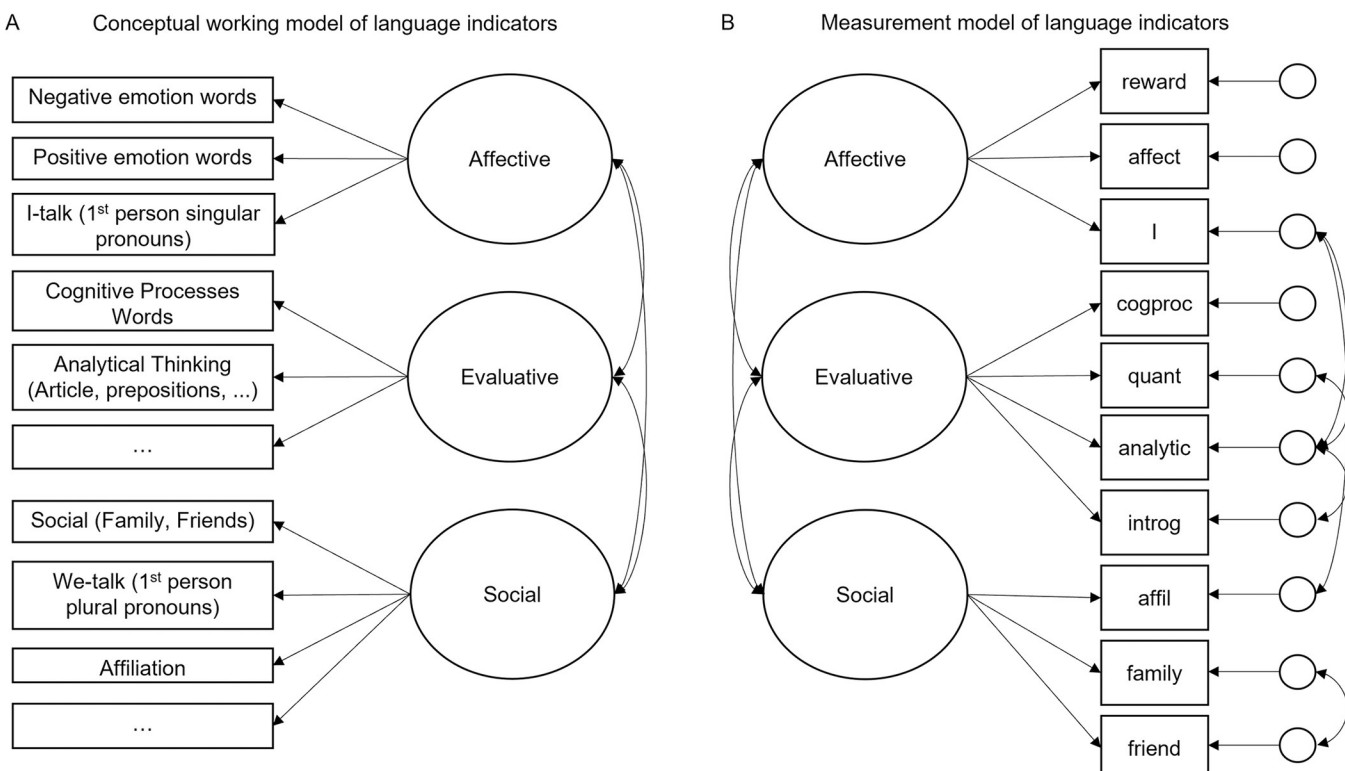

**Fig 1.** (A) Hypothesized and (B) empirically derived language markers of well-being in healthy aging narratives. (A) Conceptual working model of language indicators based on the preregistration. (B) Empirically derived measurement model of language indicators in $N = 701$ study participants. All language indicators represent word categories from Linguistic Inquiry and Word Count (LIWC). Residuals were allowed to correlate for conceptually similar language indicators or those that belonged to the same higher-order LIWC category (e.g., "friends" and "family"). Cogproc = cognitive processes, quant = quantifiers, introg = interrogatives, affil = affiliation words.

Building upon the outlined empirical findings and conceptual foundations of well-being, we integrate research in the psychology of word use in a model with the following components: (1) An affective component (which comprises the positive and negative emotions an individual tends to experience), (2) an evaluative component (which comprises the broad cognitive evaluations about life and domains of functioning, e.g., life satisfaction, subjective health), and (3) a social component (which comprises reflections underlying the perceived quality of social relationships, e.g., available social support, loneliness). Our focus is on theoretically informed, psychometrically established language markers that have extensively been used in the social sciences, namely word categories from the Linguistic Inquiry and Word Count (LIWC) [7,54]. Our approach is informed by views that see language markers as a reflection of well-being processes that might not be fully consciously accessible [20]. We suppose that language markers can be used to assess the processes that underlie and ultimately lead to awareness of the well-being dimensions assessed in self-reports. We therefore expect that the groups of language indicators for each component can be cross-validated by established self-report measures of these different well-being dimensions.

To investigate these assumptions, we use data from a large, age-diverse sample of participants who wrote personal narratives about their thoughts and feelings regarding healthy aging and completed a large set of well-being questionnaires. In a first step, we compute a measurement model of these theoretically plausible language markers to test their internal structure and identify the reliability as proposed language indicators. In a second step, we combine the language model with a set of established self-report indicators of well-being to evaluate whether the language indicators demonstrate the assumed concurrent validity with self-reported well-being dimensions.

## Materials and methods

### Sample

Data was collected as part of an online study investigating personal views on healthy aging at the University of Zurich. Participants were required to be at least 18 years old and to have native or very good writing skills in German, as the study included a writing task.

Participant recruitment and data collection took place between September 2018 and September 2020. Since the goal was to have an age-diverse sample spanning the adult life-span, data was collected in two steps. First, the link to the online study was distributed via different channels, including mailing lists, senior universities, social media, study participant pools and websites of universities, as well as local newspapers and newsletters. This yielded a preliminary sample of $N = 285$ participants, mostly comprising younger (18–30 years) and older (60+) female participants. $N = 436$ participants were then recruited via the research participant recruitment platform Prolific. The goal here was to recruit more middle-aged participants and (male) younger participants to diversify the sample. Prolific participants were eligible if German was their first language, and if they were at least 18 years old.

Out of the 721 participants who completed the study, 20 participants had to be excluded because their written essay (see "Procedure") was shorter than 100 words. By excluding these short essays, we aimed to maximize the reliability of our language measures, a common practice in LIWC research [46]. Our final sample included 701 participants (see Table 1). Our theoretical assumptions, analysis plan, and exclusion criteria were preregistered (see: https://osf.io/jwkh9). The preregistration involves research questions, analysis and hypotheses going beyond this study. Only "Research Goal I" of the preregistration is within the scope of the present work.

**Table 1. Sample description (N = 701).**

|  | *N* | % |
|---|---|---|
| Age *M (SD)* | 41.89 (19.73), Range: 18–87 | |
| Gender | | |
| Female | 341 | 48.6 |
| Male | 358 | 51.1 |
| Diverse | 1 | 0.1 |
| Marital status | | |
| Single | 408 | 58.2 |
| Married | 209 | 29.8 |
| Divorced | 60 | 8.6 |
| Widowed | 24 | 3.4 |
| Cohabiting with a partner | 292 | 41.7 |
| Children (yes) | 252 | 35.9 |
| Education (University degree or higher) | 256 | 36.5 |
| Annual income (51,000 CHF or higher) | 192 | 27.4 |
| Employment[a] | | |
| Employed | 422 | 60.2 |
| Not in gainful employment | 152 | 21.7 |
| Self-employed | 105 | 15.0 |
| Civil servants | 22 | 3.1 |
| PHQ-9 depression score *M (SD)* | 6.39 (4.84) (cut-off for mild depression = 9) | |

*Note.* 51,000 CHF (Swiss Francs) = approx. 56'000 USD.

[a]In case of retirement, this refers to the last position.

## Ethics approval and data protection

This study was conducted in accordance with the guidelines provided by the Declaration of Helsinki. All study procedures were reviewed and approved by the ethics committee of the School of Humanities and Social Sciences at the University of Zurich (approval number: 2017.10.4). Participants gave written informed consent through an online form at the beginning of the survey. The consent form contained information on (a) the study goals and procedure, (b) data protection and security (e.g., that data was stored on secured servers), (c) participants' rights regarding their data (e.g., that participation is voluntary and can be withdrawn any time) and (d) contact information. The survey only continued after consent had been provided. At the end of the survey, participants could enter their email address if they wished to be informed about future findings of the study. This was completely optional and stored separately from the study data to ensure anonymization of the data. Data was first accessed in September 2020, after data collection was completed.

## Procedure

The study was conducted on the online survey platform "Unipark". Participants first completed a 15-min writing task about their "deepest thoughts and feelings about aging well". The instructions were based off of the established expressive writing paradigm [8,15] and modified to reflect the aging context (see S1 File, Section A for the writing instructions). Participants were asked to write at least 15 minutes without interruptions. Once they started, a timer (counting down from 15 min) appeared on their screen so they could keep track of time. After the writing task, participants completed a set of questionnaires assessing different aspects of

well-being, as well as other individual differences (e.g., age stereotypes) unrelated to the present study. On average, it took participants about 1 hour to complete the study. Upon completion, participants received course credit, if eligible, and Prolific participants were compensated with 7.95 GBP (approx. 11 USD).

## Measures

**Language indicators of well-being in healthy aging narratives.** Writings of personal healthy aging views were analyzed with the German version (DE-LIWC2015) [55] of LIWC [54]. LIWC is an extensively validated and very widely used text analysis program that captures the frequencies with which a pre-defined set of word categories (e.g., emotion words) is used in a given text document. LIWC quantifies a person's word use for each category as relative percentages of the total words used by that person. For example, a person's text "I am very happy" would be scored as 25% first person singular pronouns ("I"), 25% positive emotions ("happy") and 25% quantifiers ("very").

As outlined in the preregistration, we focused on word categories that prior research had empirically or conceptually linked to well-being. Some adjustments to the pre-registration had to be made, which are outlined in "Analytical Strategy" below. An overview of all LIWC categories included in our final models is provided by Fig 1.

**Self-report indicators of well-being.** We used several well-established questionnaire scales to assess aspects of well-being such as life satisfaction, quality of life, depression, loneliness, and subjective health. Our focus was on the inclusion of relevant variables (i.e., established well-being scales), allowing us to examine links between language markers and self-reported well-being. Due to the large number of questionnaires and validity considerations, we focused on validated (sub-)scales rather than single items, so as to maximize reliability and parsimony of our models. A comprehensive overview of the scales included in our final models is provided in S1 File, Section C.

**Analytical strategy.** The first aim of this research was to establish the factor structure of language use in the context of personal healthy aging narratives. Knowing the internal structure of different language variables is a crucial first step for the development of context-sensitive language measures of well-being. In a second step, our goal was to validate the derived language factor structure by mapping language variables to our established indicators of well-being, that is, to estimate a set of latent variables of well-being measured by both language indicators and questionnaire scales. Specifically, this second step enabled us to examine the correspondence between language markers and self-reported indicators of well-being derived from conventional self-report measures, while also assessing their unique contributions.

**Establishment of the factor structure of language use in healthy aging narratives.** We conducted a series of Structural Equation Models (SEM) in Mplus to examine our research questions. As per common recommendations [56], the indices used to evaluate model fit include the Bentler's Comparative Fit Index (CFI), the Standardized Root Mean Square Residual (SRMR), and the Root Mean Square Residual of Approximation (RMSEA), for which the following values are considered as good representation of the data: $CFI > .90$, $SRMR < .08$, and $RMSEA < .08$ [56] (see "Preliminary results and model fit").

First, we tested our preregistered assumption that language indicators would map onto the three theory-based components (i.e., affective, evaluative, and social), along with a superordinate factor of overall well-being. We therefore aimed to build a measurement model in which these latent factors are measured by the preregistered observed language indicators. The preregistered measurement model, however, resulted in poor model fit, suggesting that the factor

structure we had theoretically assumed for our language variables was not well supported by the data.

For this reason, we used empirical information to modify the theoretical model and performed a series of necessary steps to eliminate potential methodological problems and reduce the number of indicators. First of all, for hierarchical categories in the LIWC dictionary (e.g., "positive emotions", which is also part of the higher-order category "affective processes"), we only included one category in our model, either the lower- or the higher-order category, in order to avoid multicollinearity and redundancies. Similarly, we removed non-significant and weakly-loading indicators and included the well-established composite score "analytical thinking" [1] instead of its various function word subcategories (personal pronouns, articles, conjunctions, etc.) to achieve a more parsimonious model.

As part of the steps taken to improve our model, we also performed a Principal Component Analysis (PCA) with varimax rotation to identify language variables, within our preregistered variables, that load onto common components. The PCA was conducted on 1/3 of our sample to avoid potential concerns of overfitting when applied in combination with confirmatory factor analysis [57]. In line with our assumptions, the PCA suggested three components of well-being, but without a higher-order factor. This procedure can be seen as a way of modelling our theoretically informed constructs by acknowledging the empirically derived structure of language variables. The reduction of the number of indicators moreover helped to avoid possible power issues. All these steps resulted in our final language measurement model (see Fig 1), which included 10 language variables loading onto three latent constructs and showed satisfactory model fit.

We allowed the residual variances to correlate for language categories that were conceptually similar or part of the same higher-order LIWC category (e.g., "family" and "friend" are both subcategories of the LIWC social processes category). Cognitive processes words had a negative, but non-significant residual variance, which we then fixed to zero without any changes in the results.

**Validation of language factor structure in healthy aging narratives.** To evaluate concurrent validity, i.e., whether the established language factor structure maps onto theoretically derived facets of well-being, we extended the language measurement model by including self-report variables of well-being. Prior to that, we wanted to make sure that the hypothesized well-being structure adequately represents the self-report data. We thus performed another SEM measurement model with the questionnaire data only. This, however, again resulted in poor model fit. We thus followed an analogous procedure as outlined above to reduce the vast number of questionnaire variables (i.e., focus on scales rather than single items, reduce indicators with non-significant or weak loadings). As part of the data reduction process, we also conducted a PCA on our preregistered self-report scales of well-being, using 1/3 of the sample. This suggested that a 2-factor structure would best represent the data, so that scales we had expected to load on affective or evaluative well-being tended to load onto one common factor. Informed by this, we then estimated a questionnaire measurement model with two latent and 30 manifest variables (i.e., self-report scales including depression, loneliness, subjective health; see S1 File, Section C for an illustration). We allowed the residual variances to correlate for conceptually similar scales that measured the same construct (e.g., life satisfaction scales).

In a final step, we tested the links between our established sets of language and self-report indicators of well-being with a path model that included both measurement models (i.e., language and self-report; see Fig 2). This can be thought of as a regression-type of analysis testing associations between the language- and questionnaire-derived latent factors. In all our models, factor variances were set to 1 and factor loadings estimated freely for model identification. All language and self-report variables had been mean-centered prior to inclusion in the models.

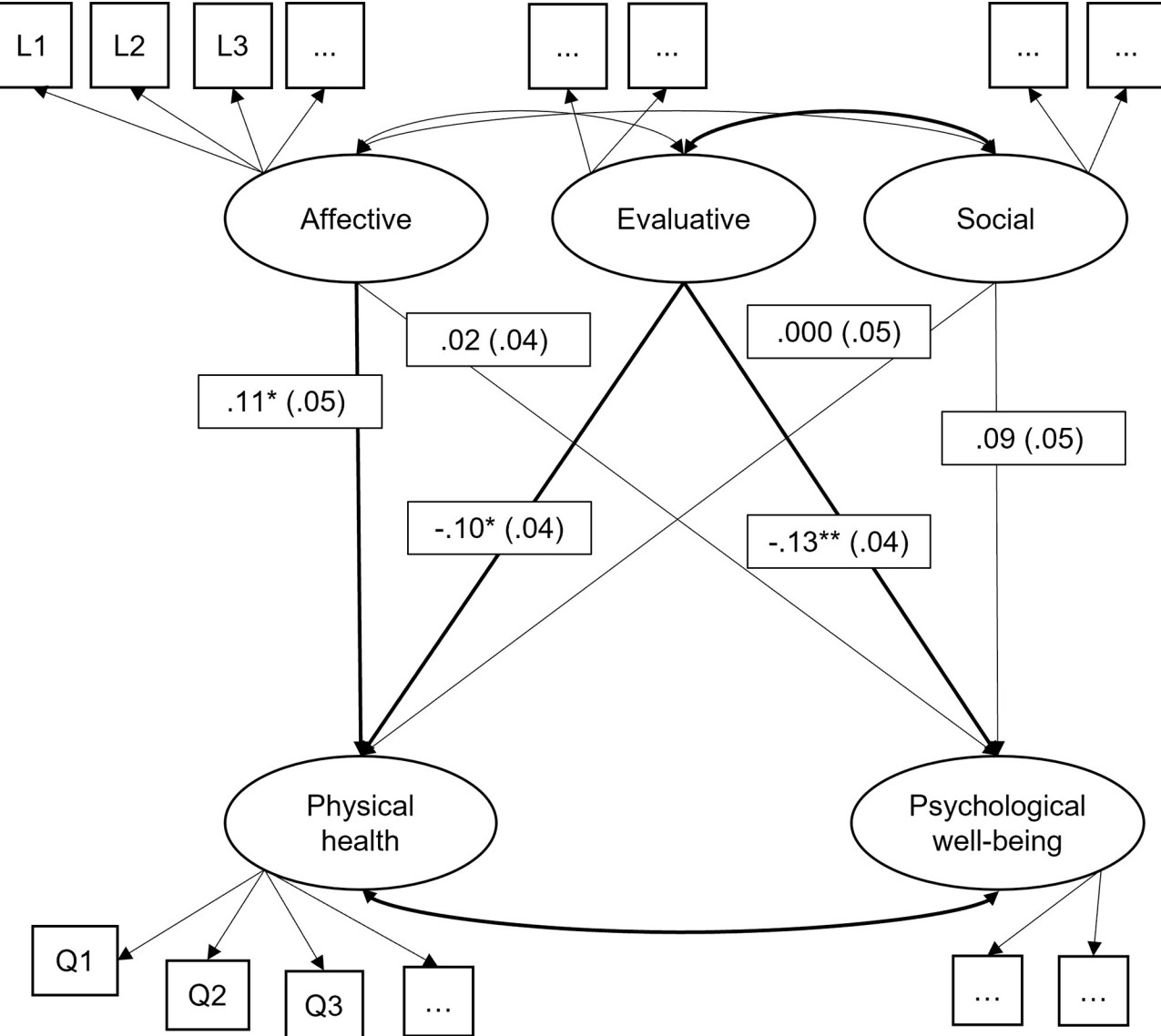

**Fig 2. Combined model with language and questionnaire indicators of well-being (schematic illustration).** *N* = 701 study participants. Depicted are standardized factor loadings with standard errors; statistically significant paths are presented in boldface. The model is only illustrated schematically for space reasons. For the full results, please refer to S1 File. Language-based latent variables: Affective, evaluative, social; questionnaire-based latent variables: Physical health, psychological well-being. L1, L2, . . . = language indicator, Q1, Q2, . . . = questionnaire indicator. * *p* < .050, ** *p* < .010, *** *p* < .001.

The data and analysis scripts used in the present article are available on the Open Science Framework: https://osf.io/s3qr6.

## Results

### Preliminary results and model fit

Means and standard deviations and intercorrelations among language variables are presented in S1 File, Section B. All models showed acceptable fit according to commonly used cut-offs [56]: Model 1 (language measurement model): χ2 (df) = 105.782 (28), CFI = 0.912, RMSEA [90% CI] = 0.063 [0.050, 0.076], SRMR = 0.044; Model 2 (questionnaire measurement model):

$\chi^2$ (df) = 1,452.827 (356), CFI = 0.930, RMSEA [90% CI] = 0.066 [0.063, 0.070], SRMR = 0.056; Model 3 (combined model with language and questionnaire indicators): $\chi^2$ (df) = 1,931.771 (678), CFI = 0.924, RMSEA [90% CI] = 0.051 [0.049, 0.054], SRMR = 0.052.

## Establishment of the factor structure of language use in healthy aging narratives

The language measurement model included 10 observed language variables that loaded on three latent factors (see Table 2). Overall, the factor structure found for the language variables seemed in line with the assumed structure, as it included latent factors related to affective processes ("affective"), evaluative processes ("evaluative"), and social processes ("social"). Counter to our expectations, however, not all latent factors were positively correlated with each other: While "affective" and "social" seemed to be independent from each other ($\beta$ = -.03, SE = .07, p = .693), "affective" and "evaluative" were positively associated with each other ($\beta$ = .12, SE = .06, p = .034), and "social" and "evaluative" were negatively associated with each other ($\beta$ = -.34, SE = .04, p < .001). This can be taken to suggest that the evaluative factor, as a language marker of cognitive-evaluative processes, may not necessarily indicate positive functioning. Within the evaluative factor, there was one language category (analytical thinking) with a negative factor loading, and three (cognitive processes words, interrogatives, and quantifiers) with positive factor loadings (see Table 2). This hints at opposing qualities within the evaluative factor with a more analytical (as opposed to narrative or dynamic) style predicting lower values, and cognitive processes words, interrogatives, and quantifiers predicting higher values in the evaluative factor. Somewhat different from our expectations, quantifier words did not load onto the latent affective dimension, but on the evaluative dimension instead. Moreover, the model fit was better when including the higher order LIWC category "affective processes" rather than its subcategories (positive and negative emotion words), suggesting that the overall emotionality of the narratives was more important than positive or negative valence. Finally, self-references (I-talk) showed the expected negative loading onto the latent affective factor. In S2 File, we provide example excerpts from healthy aging narratives that score high (vs. low) on affective, evaluative, and social language markers derived from our measurement model.

**Table 2. Standardized factor loadings in Model 1 (language measurement model).**

| Variable | Affective | Evaluative | Social |
|---|---|---|---|
| | **β [95% CI]** | | |
| Reward | 0.59*** (0.14) [0.31, 0.86] | | |
| Affect | 0.43*** (0.11) [0.22, 0.64] | | |
| I-talk (1st person singular pronouns) | -0.16** (0.05) [-0.26, -0.05] | | |
| Cognitive processes | | 1.00 (.00) [1.00, 1.00] | |
| Analytical thinking | | -0.40*** (0.03) [-0.45, -0.34] | |
| Interrogatives | | 0.24*** (0.04) [0.17, 0.31] | |
| Quantifiers | | 0.18*** (0.04) [0.11, 0.25] | |
| Affiliation | | | 0.79*** (0.05) [0.69, 0.89] |
| Family | | | 0.66*** (0.05) [0.56, 0.76] |
| Friends | | | 0.42*** (0.05) [0.33, 0.51] |

*Note.* Affective, evaluative, and social are latent variables. CI = Confidence interval. All variables had been mean-centered prior to inclusion in the model. ** p < .010, *** p < .001.

As an additional robustness check, we estimated the same measurement model separately for younger (40 years or younger), middle-aged (41–60 years) and older (60+ years) adults to see whether the observed language structure generalizes across different age groups (see S1 File, section C). Overall, the structure observed in the full sample replicated in the three age groups, indicated by similar sizes and same directions of associations compared to the main analysis. A subtle difference between age groups was observed regarding the relationship between the "affective" and "evaluative" language factors, which were no longer significantly correlated with one another (in neither of the three age groups, $p > .575$). "Affective" and "evaluative" language factors were positively (but non-significantly; $p > .575$) correlated among young and older adults (same direction as in the full sample) but were negatively (non-significantly; $p = .792$) correlated among middle-aged adults. While this hints at potential age differences in how affective and evaluative processing interact when writing about healthy aging, we advise caution in interpreting this finding due to the smaller and unequal sample sizes and limited statistical power when analyzing the three age groups separately.

## Validation of language factor structure in healthy aging narratives

Before combining language and self-report data in a single model, we computed a questionnaire measurement model to ensure an accurate representation of this data. Counter to our expectations, a model with two latent variables seemed to provide the most accurate fit of our questionnaire data (see S1 File, Section C for standardized factor loadings). One latent factor seemed to relate to subjective physical health, indicated by self-report scales on physical health, pain, or sleep quality. The other latent factor, psychological well-being, seemed to relate to a combination of affective, social, and evaluative aspects of well-being. The two latent factors (self-reported physical health and psychological well-being) were strongly correlated with each other ($\beta = .82$, SE = .02, $p = < .001$).

Finally, as a first step towards the validation of the identified language factor structure, we computed a path model combining the (a) language and (b) questionnaire measurement models (see Fig 2). Standardized factor loadings from this model are presented in S1 File, Section C. In general, language indicators accounted for 2–3.2% of the total variance in the two latent factors of self-reported physical health and psychological well-being (see Table 3). The language-based latent factor "evaluative" showed negative associations with both questionnaire-based latent factors, physical health ($\beta = -.10$, SE = .04, $p = .024$) and psychological well-being ($\beta = -.13$, SE = .04, $p = .002$; see Fig 2 and Table 3). This corroborates the assumption that the

**Table 3. Associations between language and questionnaire-based latent variables in Model 3 (combined path model).**

| Variable | β [95% CI] | | | |
| --- | --- | --- | --- | --- |
| | Psychological well-being | Physical health | *Affective* | *Evaluative* |
| *Affective* | 0.02 (0.04) n.s. [-0.06, 0.10] | 0.11* (0.05) [0.02, 0.20] | – | |
| *Evaluative* | -0.13** (0.04) [-0.21, -0.05] | -0.10* (0.04) [-0.18, -0.01] | 0.08 (0.05) n.s. [-0.02, 0.19] | – |
| *Social* | 0.09 (0.05) n.s. [-0.01, 0.18] | 0.000 (0.05) n.s. [-0.10, 0.10] | -0.02 (0.05) n.s. [-0.11, 0.07] | -0.35 (0.04) *** [-0.42, -0.27] |
| $R^2$ | .032* ($p = .023$) | .020 ($p = .107$) | | |

*Note*. Presented are standardized estimates. CI = Confidence interval. Affective, evaluative, and social are language-based latent variables (presented in italics); psychological well-being and physical health are questionnaire-based latent variables. n.s.: $p > .050$, * $p < .050$, ** $p < .010$, *** $p < .001$.

evaluative factor reflects rather maladaptive forms of cognitive-evaluative processing usually seen at lower levels of well-being. Further, the language-based factor "affective" showed a significant positive association with "physical health" (β = .11, SE = .05, $p$ = .019) but was not related to "psychological well-being" ($p$ = .666). The language-based factor "social" did not show any significant associations with self-reported well-being above and beyond the other language factors ($p$ > .050), but continued to show a negative association with "evaluative" (β = -.35, SE = .04, $p$ = < .001). A follow up analysis revealed that "social" was positively associated with "psychological well-being" ($p$ = .003) when "social" was the only language-based factor in the model, but this association was no longer statistically significant when "evaluative" was included. Regarding the manifest language indicators in the full model, I-talk no longer showed a significant association with the affective factor above and beyond all other variables ($p$ = .088).

In conclusion, the combined path model suggests that the structure of language use with three latent factors ("affective", "evaluative", and "social"), at least partially, reflects self-reported well-being. The partial contribution of the "evaluative" language factor seemed to reflect maladaptive processes as it was related with poorer self-reported well-being outcomes.

## Discussion

The present study used personal narratives about healthy aging in an adult lifespan sample and computerized text analysis to systematically identify linguistic indicators of well-being. Based on previous research, we proposed a conceptual model of language-based, candidate indicators of well-being and presented empirical evidence towards a first validation of this model in the context of healthy aging narratives.

In a two-step approach, we first fitted a measurement model, which confirmed the hypothesized structure of language indicators pointing to three latent factors (reflecting affective, evaluative, and social processes). As one of the first empirical examinations of the internal structure of language indicators of well-being, this yields new insights into the multi-faceted nature of word use and may inform future developments of language measures of well-being and related processes. There were small-to-moderate associations between the three language components, except for the "affective" and "social" components which were independent from each other. This suggests that while the three language components formed a coherent structure together, they mostly seemed to capture unique processes.

Notably, the language structure differed from the one we found for self-reported well-being scales, which was best described by two correlated factors (i.e., physical health and psychological well-being). The aggregation of various theoretically assumed factors into a single latent psychological well-being factor in self-reports echoes previous work in this area [26,58]. The differing structure observed for language compared to self-report indicators converges with theoretical perspectives [20] on language markers as a reflection of non-conscious processes, which may differ from the consciously accessible aspects of well-being typically assessed by self-reports. It also converges with other multi-method studies [5], where, in the absence of shared method variance (i.e., when the variance is attributable to the measurement method rather than to the constructs the measures are assumed to represent); [59,60]) the remaining explaining variance tends to be smaller. Language and self-report indicators may thus capture overlapping, but non-identical aspects of well-being. To test concurrent validity with established indicators of well-being, we combined language and self-report indicators ($N$ = 30 well-being subscales) in a single model, which showed various associations between language and self-reported dimensions of well-being. Our results corroborate the assumption that language patterns may reflect relevant processes of well-being and that the two measurement levels, language use and self-report, express related, yet not identical facets of well-being.

Interestingly, the structure of language indicators replicated across different age groups (young, middle, older adulthood), converging with other studies [12]. Potential subtle age differences were also observed (especially for middle-aged adults in affective and evaluative language use), reminiscent of other documented differences between middle-aged and older adults when reflecting on aging [13]. We advise caution in interpreting findings from this supplemental analysis due to smaller samples and reduced power. Future longitudinal studies should examine how views about healthy aging change within individuals as they get older and time horizons shift [13], especially since views on aging may affect actual aging [14]. Our analyses represent a first empirical validation of a language-based well-being model in the context of healthy aging that warrants further replication and validation. In the following, we discuss the results in more detail.

## Language markers of well-being in healthy aging narratives: Adaptive and maladaptive ways of thought processing

Our model supports the presence of language markers that reflect an evaluative component of well-being. A group of language indicators (i.e., cognitive processes words, interrogatives, quantifiers, and analytical thinking) of cognitive elaboration was related to self-reported physical health and psychological well-being. While this speaks for the assumed connection between language markers of cognitive elaboration and well-being, there was also an important nuance in this pattern, as the evaluative language factor seemed to reflect less adaptive processes than expected. Although we would have expected a cognitively engaging thinking style in healthy aging narratives to reflect adaptive processes of well-being, the empirically found evaluative language factor was negatively associated with self-reported physical health and psychological well-being as well as with other language dimensions (e.g., social). The evaluative language factor may thus indicate important, but rather maladaptive processes, at least in the context of healthy-aging narratives. This contributes to a nuanced conceptualization of cognitive elaboration markers.

Interestingly, the evaluative language factor also seemed to represent two distinct styles of cognitive elaboration (see S2 File for examples): Three language indicators (i.e., cognitive processes words, interrogatives, quantifiers) loaded positively onto the evaluative language factor, but analytical thinking style loaded negatively. In other words, when conjointly analyzed, the use of cognitive processes words, interrogatives and quantifiers was linked with poorer self-reported well-being, and analytical thinking style with higher self-reported well-being. This is in line with other findings showing that cognitive processes words may not always reflect desirable processes, but sometimes maladaptive processes like ruminative brooding [45]. Particularly when writing about healthy aging and anticipating the own aging process, the use of a lot of cognitive processes words, especially in combination with interrogative (e.g., why? when?) and quantifier (e.g., few, many, very) words, might indicate tendencies to ruminate and being preoccupied with negative aspects of aging. This converges with other findings on ruminative processing that revealed repeated references to "why" and "what if?" as indicators of maladaptive processing of emotionalizing content (e.g., in psychotherapy) [61,62].

Our results can further be taken to suggest that writing about healthy aging in a more analytical style (analytical thinking style was negatively associated with the maladaptive evaluative factor) rather than in a personally involved, dynamic style reflects functional processes related to greater well-being – possibly by combining cognitive elaboration with psychological distancing [44]. Analytical thinking style might reflect beneficial thought processing, and it has been shown that this might be particularly the case in less controllable contexts that require palliative coping and emotion regulation (e.g., the Covid-19 pandemic; [63]). Our study asked

participants to write about healthy aging, which may have prompted them to also confront themselves with potentially threatening prospects of getting older (e.g., physical impairments). Individuals who process their thoughts about aging in a more analytical and less personal way seemed to be better off, indicated by higher self-reported well-being. In general, it may depend on the context whether language markers of cognitive engagement and analytical thinking reflect maladaptive or more constructive forms of self-reflection. Context-dependencies have been identified for other language markers (e.g., I-talk) [39], and future research should examine whether the link between cognitive elaboration markers and well-being varies by context (e.g., in personal narratives that are not about aging).

### Emotion references relate to greater physical health, social references relate to greater psychological well-being

Beyond an evaluative language factor, an affective language factor (i.e., emotion words, reward words, I-talk) was also shown to reflect self-reported well-being. Specifically, the language-based affective factor was linked to greater physical health, converging with several studies that showed health benefits of expressing emotions [15,64]. The ability to put feelings into words has been seen as an implicit form of emotion regulation with beneficial effects at the experiential, autonomic, and behavioral level [64]. Despite its positive associations with self-reported well-being in our model, the affective language factor also showed positive associations with the maladaptive evaluative language factor. This implies that emotional expression might not always reflect adaptive processes, which has also been documented in other studies [65].

In general, our results point to a need for a broad understanding of language markers of affective well-being that goes beyond mere emotion labels and includes affective processes, self-focus (I-talk), and words referring to rewarding, emotional processes (e.g., fulfilling, rewarding, win, succeed). Our measurement model showed a better fit when including emotion words as a general category, rather than separate positive and negative emotion word subcategories. This may suggest emotional expression as a basic process involved in dealing with positive and negative emotional responses when writing about healthy aging. Future research should extend our work by examining the role of specific emotions, including self-transcendent emotions (e.g., gratitude, hope, awe) [66]. Self-transcendent emotions, such as gratitude, have consistently been associated with higher well-being across adulthood [67,68]. Future studies should explore whether individuals who focus on themes of gratitude, hope, or awe when reflecting on healthy aging experience greater well-being.

In line with prior work [5], self-references (I-talk) were part of the affective language factor, but the associations were less clear than expected as they failed to reach statistical significance in the final, combined model with language and self-report indicators. This suggests that explicit emotion labelling, indicated by emotion and reward words, may represent more relevant processes for well-being in this context. Although I-talk has been seen as a consistent marker of negative emotionality [5], this has recently been questioned by evidence suggesting that self-referential language can have both positive and negative qualities [4,39]. Especially in the aging context, I-talk might sometimes reflect wise self-reflection [4] and potential moderators could have obscured associations with well-being.

In line with our expectations, we found a social component of language use that included references to social integration (i.e., friends, family, affiliation). The language-based social factor was linked with greater self-reported psychological well-being, indicating concurrent validity. This is not surprising, given the crucial role social integration plays for well-being and healthy aging [29,30,32]. This association, however, failed to reach statistical significance once the other latent language-based factors were included in the model. Robust negative

relationships between the social and evaluative language factors moreover suggest that individuals who reflected more on being socially integrated in their writings about aging may be less likely to engage in rumination and other maladaptive thinking styles (represented by the evaluative factor).

In conclusion, we identified three language factors that were meaningfully related to each other and to self-reported well-being. The distinct association patterns between each language factor and self-reported well-being allude to the uniqueness of each of the three language factors, despite some overlap. The evaluative language dimension showed the most consistent (but negative) associations with self-reported well-being. Besides the expected associations of language indicators with established self-report scales, which speak to concurrent validity, there were also unexpected ones. Our findings reveal different features of cognitive processing that reflect adaptive and maladaptive thought processing. They invite for new conceptualizations and further testing of cognitive elaboration markers as indicators of well-being and underlying risk factors.

## Limitations and outlook

The present study has several limitations, which we would like to point out along with recommendations for future research. Although we were able to recruit an age-diverse sample (18–87 years), (a) certain demographic groups (e.g., middle-aged men) were underrepresented in our study. Further replication of our results in diverse study populations (e.g., pooled study samples) are needed before drawing premature conclusions. In light of the documented socioeconomic [69] and racial [70] health disparities in aging, replication in socioeconomically, ethnically, and culturally diverse samples will be important. It will also be important (b) to probe the generalizability of the findings across different sources and contexts of language data (e.g., diary entries). Beyond further replication, our study opens the door for (c) longitudinal studies that will offer more fine-grained insights into the within-person dynamics of language and well-being over time. Longitudinal studies will also allow to control for possible cohort effects [71] and between-person traits (e.g., personality) that could affect how language relates to how an individual ages over time. Lastly, (d) the effect sizes were small. Only between 2–3.2% of the total variance in self-reported well-being were accounted for by the language model, in line with other research [5]. It needs to be considered, however, that there is no shared method variance and that associations between behavioral and self-report data tend to be small in spite of being relevant [59,72]. In fact, promising evidence shows that language analysis can complement (or even outperform) self-report questionnaires in predicting real-life health outcomes [20]. This alludes to the potentially unique perspective provided by language markers as opposed to self-report measures, and future research will not only benefit from refining relevant language markers, but eventually also from combining language with other behavioral markers of well-being. In the future, relevant language dimensions could be used to develop automated feedback tools that enable the monitoring of individuals' well-being based on their language (e.g., in diaries) [73]. Our results might inform the burgeoning field of digital health interventions [74], which often uses language alongside other automatically sensed behavioral markers to unobtrusively monitor well-being.

## Conclusion

In the present study, we identified a conceptual language model that integrates different facets of well-being and demonstrated the internal structure of language indicators (i.e., affective, evaluative, and social components) in the context of healthy aging narratives. The study provides first evidence of their concurrent validity as reliable markers of core components of well-

being and underlying processes. While there are many different approaches to natural language processing (with varying degrees of complexity)[37], the meaning and validity of automatically extracted features from large models often remains unclear. This study aims to contribute to a deeper understanding of the psychometrics of natural language use. Innovative and more implicit measures of well-being are urgently needed, and language data continues to become more accessible in the digital era. In the future, once language indicators have been more thoroughly validated as markers of well-being, this will eventually allow their extraction from the language people naturally produce (e.g., online or in everyday speech) to infer individuals' well-being at scale. We hope that this work can contribute to the development of more and more innovative behavioral measures that will eventually complement self-reports and help us to better understand and assess well-being and healthy aging.

## Supporting information

**S1 File. Additional information on study materials and detailed overview of results.**
(DOCX)

**S2 File. Example excerpts from healthy aging narratives.**
(DOCX)

**S3 File. STROBE checklist.**
(DOCX)

## Acknowledgments

We would like to thank Zilla M. Huber for her valuable contributions to data collection. We sincerely thank all the study participants. Tabea Meier and Andrea B. Horn are affiliated with, and Mike Martin is the director of the URPP "Dynamics of Healthy Aging". During her work on this project, Tabea Meier was a pre-doctoral fellow of LIFE (International Max Planck Research School on the Life Course; participating institutions: MPI for Human Development, Humboldt-Universität zu Berlin, Freie Universität Berlin, University of Michigan, University of Virginia, University of Zurich).

## Author Contributions

**Conceptualization:** Tabea Meier, Matthias R. Mehl, Andrea B. Horn.

**Data curation:** Tabea Meier.

**Formal analysis:** Tabea Meier, Andrea B. Horn.

**Funding acquisition:** Mike Martin.

**Investigation:** Tabea Meier, Matthias R. Mehl, Mike Martin, Andrea B. Horn.

**Methodology:** Tabea Meier, Matthias R. Mehl, Andrea B. Horn.

**Project administration:** Tabea Meier.

**Resources:** Matthias R. Mehl, Mike Martin, Andrea B. Horn.

**Supervision:** Matthias R. Mehl, Mike Martin, Andrea B. Horn.

**Visualization:** Tabea Meier.

**Writing – original draft:** Tabea Meier, Andrea B. Horn.

**Writing – review & editing:** Tabea Meier, Matthias R. Mehl, Mike Martin, Andrea B. Horn.

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
