## [Decision Letter · Decision Letter 0]

21 Aug 2023

PONE-D-23-15452When I am Sixty-Four… Evaluating Language Markers of Well-Being in Healthy Aging NarrativesPLOS ONE

Dear Dr. Meier,

Thank you for submitting your manuscript to PLOS ONE. After careful consideration, we feel that it has merit but does not fully meet PLOS ONE’s publication criteria as it currently stands. Therefore, we invite you to submit a revised version of the manuscript that addresses the points raised during the review process.

We look forward to receiving your revised manuscript.

Kind regards,

Michal Ptaszynski, PhD

Academic Editor

PLOS ONE

Journal Requirements:

Reviewers' comments:

Reviewer's Responses to Questions

**Comments to the Author**

1. Is the manuscript technically sound, and do the data support the conclusions?

Reviewer #1: Partly

Reviewer #2: Yes

2. Has the statistical analysis been performed appropriately and rigorously? 

Reviewer #1: Yes

Reviewer #2: Yes

3. Have the authors made all data underlying the findings in their manuscript fully available?

Reviewer #1: Yes

Reviewer #2: Yes

4. Is the manuscript presented in an intelligible fashion and written in standard English?

Reviewer #1: Yes

Reviewer #2: Yes

5. Review Comments to the Author

Reviewer #1: Using online surveys and computerized textual analysis, this manuscript presents research correlating the linguistic features of an adult sample's Healthy Aging Narratives with their self-reported well-being measures. Overall, I find the manuscript to be well-written and easy to follow. The end goal of constructing a set of word-based measures for the analysis of population well-being is sound and meaningful, with the results presented cleanly. Below, I provide three main points of consideration, along with a few minor comments.

Firstly, the study uses a German adult sample encompassing individuals across all age cohorts, which might require further justification: Given that the primary context of the research pertains to “aging,” wouldn't focusing solely on older adults be more appropriate? Studies such as Twenge et al., 2012 have demonstrated that young adults often have a distinct set of priorities that differs from older adults, changing as they age. Therefore, it seems more logically consistent to base the project on “happy” older adults. Although the authors might have more substantial reasons, it’s crucial to distinguish between “expected” healthy aging and actual healthy aging.

Related, as the study focuses on healthy aging, some key research on this subject, such as the HANDLS (https://handls.nih.gov/) and the Chicago Health and Aging Project (CHAP) appear to be omitted. It’s important to draw appropriate lessons from these pioneering and longitudinal studies.

My second point concerns the model the authors built for testing. Specifically, the three-factor language model calls for a more robust theoretical foundation. For instance, the Social Component was identified as a primary factor, but it could also be argued that social relationships might be an exogenous factor leading to affective well-being. In terms of affective well-being, research in positive psychology, particularly on self-transcendent emotions, suggests other-oriented emotions are highly relevant to one’s well-being. These should be considered, if not specifically examined (using LIWC) in this project. Overall, the authors need to provide stronger justifications for their choice of language markers.

Thirdly, in the final stage of data analysis, where the combined model with language and questionnaire indicators of well-being was tested, the authors did not appear to include any self-reported trait factors as covariates. This omission could raise theoretical concerns. One might envision a scenario where a highly extroverted person scores themselves highly in the survey and expresses joy through many positive words (and vice versa). In such cases, trait factors like the big five should be included as covariates. I understand that the authors may not have measured the big five explicitly, but measures such as “mental health” and “loneliness” could be included in the model (as proxies for trait factors).

Minor points:

1. lines 441-442, I may not be familiar with the terms used, but what do “shared method variance” and “explaining variance” mean?

2. Figures 1-3 appear to be auto-generated by Mplus. Enhancing the graphic quality of these figures could improve reader comprehension of the model. For instance, the legibility of the coefficients in Figure 3 is somewhat low.

3. On line 298, I'm assuming a CFA was conducted? The authors mentioned poor model fit, and some readers might want to see the fit indices for the model. Also, since “PCA was conducted on 1/3 of our sample,” please specify the sample size used in each model within your figures.

Twenge, J. M., & Campbell, S. M. (2012). Who are the Millennials? Empirical evidence for generational differences in work values, attitudes and personality. Managing the new workforce: International perspectives on the millennial generation, 152-180.

Reviewer #2: This is a very nice paper, well written and with very detailed analyses.

Although I consider that nowadays methods like LIWC are not considered state of the art given the recent boom of large language models (e.g., the use of transformers where context is considered), I think that the level of analysis performed is very interesting to be shared in the community.

Here are some minor comments that may help improve the manuscript.

1. When I read the title I thought that the analysis was focused on age, but the question is only related to aging. I wonder if people 18 and older have the same perception of aging as people older than 70. Maybe using “When I am 64…” is misleading. Did the authors check if by thresholding by age, results changed?

2.I think it would be nice to present some examples actually showing how people answer to these questions. Even some sentences in which some of the words had been considered in the LIWC features.

3.In addition, I consider that maybe you can merge Fig 1 & 2 and include a table of results from the supplemental into the main manuscript, which are very informative.

4.I was also impressed but the number of self-assessment tests that you integrated for the analysis. I think it needs to be highlighted in the main manuscript, especially in the abstract.

6. PLOS authors have the option to publish the peer review history of their article (what does this mean?). If published, this will include your full peer review and any attached files.

Reviewer #1: No

Reviewer #2: No

---

## [Author Response · Author response to Decision Letter 0]

5 Oct 2023

(Please see attached file 'Response to Reviewers')

Dear Dr. Meier,

Thank you for submitting your manuscript to PLOS ONE. After careful consideration, we feel that it has merit but does not fully meet PLOS ONE’s publication criteria as it currently stands. Therefore, we invite you to submit a revised version of the manuscript that addresses the points raised during the review process.

We look forward to receiving your revised manuscript.

Kind regards,

Michal Ptaszynski, PhD

Academic Editor

PLOS ONE

Dear Dr. Ptaszynski,

We appreciate the opportunity to revise our manuscript entitled "When I am Sixty-Four… Evaluating Language Markers of Well-Being in Healthy Aging Narratives" and would like to thank the reviewers for their helpful feedback and suggestions.

We carefully addressed each reviewer comment and respond to each point below. We believe that the manuscript has benefited from the reviewer comments and the revision, and we hope you and the reviewers agree.

Thank you so much for considering our revised manuscript.

Response to Review Comments

(Page numbers refer to the marked-up copy of the manuscript)

Reviewer #1: Using online surveys and computerized textual analysis, this manuscript presents research correlating the linguistic features of an adult sample's Healthy Aging Narratives with their self-reported well-being measures. Overall, I find the manuscript to be well-written and easy to follow. The end goal of constructing a set of word-based measures for the analysis of population well-being is sound and meaningful, with the results presented cleanly. Below, I provide three main points of consideration, along with a few minor comments.

Response: Thank you for the positive feedback! We appreciate the valuable suggestions, which helped us improve the manuscript substantially. 

Firstly, the study uses a German adult sample encompassing individuals across all age cohorts, which might require further justification: Given that the primary context of the research pertains to “aging,” wouldn't focusing solely on older adults be more appropriate? Studies such as Twenge et al., 2012 have demonstrated that young adults often have a distinct set of priorities that differs from older adults, changing as they age. Therefore, it seems more logically consistent to base the project on “happy” older adults. Although the authors might have more substantial reasons, it’s crucial to distinguish between “expected” healthy aging and actual healthy aging.

Response: Thank for raising this important point and for the opportunity to clarify our theoretical approach. Our study is based on the WHO (2015) definition that postulates healthy aging as a lifelong process that is not defined by a certain age but rather as a shared human experience.

Therefore, we made sure that our language samples were prompted by the mental representations, thoughts and feelings regarding aging well as process. We expect that writing about aging is a suitable context to activate the processes of interest, which in turn will manifest in language markers of current well-being.

We now provide a clearer rationale for our lifespan approach in the introduction and have added needed elaboration and additional citations (see p. 3 + 4). 

Our approach is in line with earlier findings from qualitative studies showing that individuals across all ages do reason and feel about aging, and they do so in broadly comparable ways (Jopp et al., 2015). Another line of research highlights the importance of subjective views on aging at younger ages, as views and expectations of aging can turn into a self-fulfilling prophecy when people get older and thereby impact “actual” aging (e.g., Levy et al., 2002). We now elaborate more on this in the discussion section (p. 25, 31) and refer to relevant literature and future directions in this area. 

Yet, we wanted to acknowledge that views on aging and aging well might also change as people get older. Longitudinal designs that allow to control for cohort effects (see e.g., Twenge et al., 2012) will be needed to properly investigate this. The paradigm with automated language analysis offers a scalable solution to investigate language samples, and the study opens the door for more fine-grained and longitudinal studies in the future to fill this gap in the literature. We elaborate on these thoughts in the limitations & future directions section (p. 31). 

Jopp, D. S., Wozniak, D., Damarin, A. K., De Feo, M., Jung, S., & Jeswani, S. (2015). How could lay perspectives on successful aging complement scientific theory? Findings from a US and a German life-span sample. The Gerontologist, 55(1), 91-106.

Levy, B. R., Slade, M. D., Kunkel, S. R., & Kasl, S. V. (2002). Longevity increased by positive self-perceptions of aging. Journal of Personality and Social Psychology, 83(2), 261–270. https://doi.org/10.1037/0022-3514.83.2.261

Twenge, J. M., & Campbell, S. M. (2012). Who are the Millennials? Empirical evidence for generational differences in work values, attitudes and personality. Managing the new workforce: International perspectives on the millennial generation, 152-180.

Related, as the study focuses on healthy aging, some key research on this subject, such as the HANDLS (https://handls.nih.gov/) and the Chicago Health and Aging Project (CHAP) appear to be omitted. It’s important to draw appropriate lessons from these pioneering and longitudinal studies.

Response: Thank you for pointing us to these relevant pieces of literature. We have now incorporated additional references in the literature review of the manuscript (see p. 6), embedding our study within this line research. We further discuss our findings referring to findings of these pioneering studies (p. 30). 

My second point concerns the model the authors built for testing. Specifically, the three-factor language model calls for a more robust theoretical foundation. For instance, the Social Component was identified as a primary factor, but it could also be argued that social relationships might be an exogenous factor leading to affective well-being. In terms of affective well-being, research in positive psychology, particularly on self-transcendent emotions, suggests other-oriented emotions are highly relevant to one’s well-being. These should be considered, if not specifically examined (using LIWC) in this project. Overall, the authors need to provide stronger justifications for their choice of language markers.

Response: 

Thank you for the opportunity to clarify. Our language model is informed by the literature in the field of psychological language analysis (e.g., LIWC) and the psychometrics of language indicators. Different lines of research featured the validity of language style markers which have conceptually been linked to well-being: Affective language (e.g., Tackman et al., 2019), cognitive-analytical language (e.g., Pennebaker et al., 1997, 2014; Kleim et al., 2018), and social language (e.g., Karan et al., 2019; Rohrbaugh et al., 2008) were identified as candidate variables. Informed by these pioneering studies, our study combines these language markers in a single model to test empirical associations with self-reported well-being dimensions. This converges with conceptual views on a social component of well-being that is distinguishable from affective well-being (e.g., Ryff et al., 1989). We do acknowledge that socio-affective processes are highly interrelated. We made needed changes to the manuscript and reworded the corresponding sections in the introduction (see p. 4, 6). 

Unfortunately, we do not have the possibility to separately test self-transcendent emotion words in our pre-registered analysis, but we do pick up on this idea as an interesting future direction (see p. 29).

Tackman, A. M., Sbarra, D. A., Carey, A. L., Donnellan, M. B., Horn, A. B., Holtzman, N. S., ... & Mehl, M. R. (2019). Depression, negative emotionality, and self-referential language: A multi-lab, multi-measure, and multi-language-task research synthesis. Journal of personality and social psychology, 116(5), 817-834.

Pennebaker, J. W. (1997). Writing about emotional experiences as a therapeutic process. Psychological science, 8(3), 162-166.

Pennebaker, J. W., Chung, C. K., Frazee, J., Lavergne, G. M., & Beaver, D. I. (2014). When small words foretell academic success: The case of college admissions essays. PloS one, 9(12), e115844.

Kleim, B., Horn, A. B., Kraehenmann, R., Mehl, M. R., & Ehlers, A. (2018). Early linguistic markers of trauma-specific processing predict post-trauma adjustment. Frontiers in Psychiatry, 9, Article 645. https://doi.org/10.3389/fpsyt.2018.00645

Karan, A., Rosenthal, R., & Robbins, M. L. (2019). Meta-analytic evidence that we-talk predicts relationship and personal functioning in romantic couples. Journal of Social and Personal Relationships, 36(9), 2624-2651.

Rohrbaugh, M. J., Mehl, M. R., Shoham, V., Reilly, E. S., & Ewy, G. A. (2008). Prognostic significance of spouse we talk in couples coping with heart failure. Journal of Consulting and Clinical Psychology, 76(5), 781–789. https://doi.org/10.1037/a0013238

Thirdly, in the final stage of data analysis, where the combined model with language and questionnaire indicators of well-being was tested, the authors did not appear to include any self-reported trait factors as covariates. This omission could raise theoretical concerns. One might envision a scenario where a highly extroverted person scores themselves highly in the survey and expresses joy through many positive words (and vice versa). In such cases, trait factors like the big five should be included as covariates. I understand that the authors may not have measured the big five explicitly, but measures such as “mental health” and “loneliness” could be included in the model (as proxies for trait factors).

Response: 

This is a valid point. The beauty of language is that, almost by definition, the words people use reflect their thoughts, feelings, experiences, and traits (e.g., Tausczik et al., 2010). In our study, we opted for the most parsimonious model considering the already high number of parameters (all self-report indicators of well-being are being accounted for in our model) and the between-subjects design of our analysis. Partialling out certain additional traits like personality would obscure the associations between language and well-being markers, for example, extraversion and affect and social experiences are highly interrelated, which would limit the variance left to explain and make it harder to interpret. 

We do acknowledge, however, that controlling for between-person traits (e.g., personality) would be crucial in longitudinal studies with a focus on within-person dynamics. We refer to that in the discussion (see p. 31). 

Tausczik, Y. R., & Pennebaker, J. W. (2010). The psychological meaning of words: LIWC and computerized text analysis methods. Journal of language and social psychology, 29(1), 24-54.

Minor points:

1. lines 441-442, I may not be familiar with the terms used, but what do “shared method variance” and “explaining variance” mean?

Response: 

By explaining variance, we mean that the predictors account for meaningful variation in the response variables while there are always other factors that are not captured (as is common in this kind of research; see e.g., Matz et al., 2017). In language analysis, this is often due to lack of common (or shared) variance of measurement between predictor (e.g., a language measure) and outcome (e.g., a self-report scale). Shared method variance here refers to cases when the variance is attributable to the overlapping measurement method (e.g., self-report) rather than to the constructs the measures are assumed to represent (see e.g., Podsakoff et al., 2003; Matz et al., 2017). 

To clarify, we have now reworded the corresponding sections (p. 25) and added additional citations for the readers’ reference. 

Podsakoff, P. M., MacKenzie, S. B., Lee, J.-Y., & Podsakoff, N. P. (2003). Common method biases in behavioral research: A critical review of the literature and recommended remedies. Journal of Applied Psychology, 88(5), 879–903. https://doi.org/10.1037/0021-9010.88.5.879

Matz, S. C., Gladstone, J. J., & Stillwell, D. (2017). In a world of big data, small effects can still matter: A reply to Boyce, Daly, Hounkpatin, and Wood (2017). Psychological science, 28(4), 547-550.

2. Figures 1-3 appear to be auto-generated by Mplus. Enhancing the graphic quality of these figures could improve reader comprehension of the model. For instance, the legibility of the coefficients in Figure 3 is somewhat low.

Response: We agree and have made necessary changes to the figures. Specifically, all figures have been prepared using the tool “PACE” to ensure that they meet PLOS’ quality requirements. (We note that the resolution may still appear lower in the compiled PDF, but should be higher when downloading the figure files). 

3. On line 298, I'm assuming a CFA was conducted? The authors mentioned poor model fit, and some readers might want to see the fit indices for the model. Also, since “PCA was conducted on 1/3 of our sample,” please specify the sample size used in each model within your figures.

Response: Following this suggestion, we have added the sample size to each figure caption. We mention the model fit indices on p. 19. To ensure ease of retrieval for readers, we have added a header (“Preliminary results and model fit”) to this section, right at the beginning of the results. We refer to this section in the method section when model fit considerations are first mentioned (p. 16).

Twenge, J. M., & Campbell, S. M. (2012). Who are the Millennials? Empirical evidence for generational differences in work values, attitudes and personality. Managing the new workforce: International perspectives on the millennial generation, 152-180.

 

Reviewer #2: This is a very nice paper, well written and with very detailed analyses.

Although I consider that nowadays methods like LIWC are not considered state of the art given the recent boom of large language models (e.g., the use of transformers where context is considered), I think that the level of analysis performed is very interesting to be shared in the community.

Response: Thank you for the positive feedback! We are glad you liked our manuscript and are grateful for the suggestions on how to further improve it. 

Here are some minor comments that may help improve the manuscript.

1. When I read the title I thought that the analysis was focused on age, but the question is only related to aging. I wonder if people 18 and older have the same perception of aging as people older than 70. Maybe using “When I am 64…” is misleading. Did the authors check if by thresholding by age, results changed?

Response: Following your suggestion, we reran our language measurement model separately for different age groups. The sample was not big enough to run multi-group comparisons, nor to run the combined model (which includes a high number of language + self-report indicators) separately for different age groups, i.e., the models did not converge. Since the focus of this study is on the structure of the language indicators, we reran the more parsimonious language measurement model separately for each age group, i.e., one for young (40 years or younger), middle-aged (40-60 years), and older (60+ years) adults. 

These supplemental analyses showed that the pattern of the findings remained robust across different age groups with different significance levels due to the smaller sample size and power, but comparable sizes and same directions of the effects observed in the full sample. When analyzing the three age groups separately, the “evaluative” and “social” language components were still negatively and significantly correlated with each other, thus replicating the findings observed in the full sample. Overall, this suggests that the structure of language indicators generalizes across different age groups, which is in line with other studies that relied on lifespan samples to assess attitudes and views on aging (e.g., Jopp et al., 2015). 

The subgroup analysis also revealed potential subtle age differences regarding the relationship between the “affective” and “evaluative” language components, which were no longer significantly (p > .005) correlated with one another (in neither of the three age groups) when analyzing the three age groups separately. Like in the full sample, the “affective” and “evaluative” language components continued to be positively (but non-significantly, p > .575) correlated with each other in the young and older adults subsample but were negatively (non-significantly, p = .792) correlated with each other in the middle-aged subsample. While this hints at potential age differences in how affective and evaluative processing relate to one another when writing about heathy aging, we advise caution in interpreting this result due to the small and unequal sample sizes and reduced statistical power when analyzing the three age groups separately.

We added these analyses to the supplemental material (S1 File) and refer to them in the main manuscript on p. 21.

We further acknowledge in the discussion section that ideas of "aging well" might change during the life course, as has also been pointed out by reviewer #1. We now discuss this in the discussion section (p. 25, 31) and point to the literature as well as relevant future directions of this work. Thank you for raising this relevant point.

Jopp, D. S., Wozniak, D., Damarin, A. K., De Feo, M., Jung, S., & Jeswani, S. (2015). How could lay perspectives on successful aging complement scientific theory? Findings from a US and a German life-span sample. The Gerontologist, 55(1), 91-106.

2.I think it would be nice to present some examples actually showing how people answer to these questions. Even some sentences in which some of the words had been considered in the LIWC features.

Response: Thank you for suggesting this; we agree that this serves as a helpful illustration of our findings. We now highlight a few examples of texts that scored high (vs. low) in the examined affective, evaluative, and social language markers in a supplemental table (see S2 File), and refer to those examples in the main manuscript (e.g., p. 21, 26). (We only included example writings of participants who had agreed to the sharing of their anonymized writings).

3.In addition, I consider that maybe you can merge Fig 1 & 2 and include a table of results from the supplemental into the main manuscript, which are very informative.

Response: Given the rather high numbers of figures/tables, we were indeed a bit unsure ourselves on how to best present our findings. Following your suggestion, we have now moved one table (Table 3) from the Supplement to the main manuscript, and merged Figure 1 & 2 into one single figure. Thank you for suggesting this. 

4.I was also impressed but the number of self-assessment tests that you integrated for the analysis. I think it needs to be highlighted in the main manuscript, especially in the abstract.

Response: Thank you. We now mention the number of self-report questionnaires in the abstract. Additionally, we now mention this strength in the discussion section (p. 25) of the manuscript.

We would like to reiterate our appreciation for both reviewers’ valuable feedback!

---

## [Decision Letter · Decision Letter 1]

6 Feb 2024

PONE-D-23-15452R1When I am Sixty-Four… Evaluating language markers of well-being in healthy aging narrativesPLOS ONE

Dear Dr. Meier,

Thank you for submitting your manuscript to PLOS ONE. After careful consideration, we feel that it has merit but does not fully meet PLOS ONE’s publication criteria as it currently stands. Therefore, we invite you to submit a revised version of the manuscript that addresses the points raised during the review process.

We look forward to receiving your revised manuscript.

Kind regards,

Michal Ptaszynski, PhD

Academic Editor

PLOS ONE

Reviewers' comments:

Reviewer's Responses to Questions

**Comments to the Author**

1. If the authors have adequately addressed your comments raised in a previous round of review and you feel that this manuscript is now acceptable for publication, you may indicate that here to bypass the “Comments to the Author” section, enter your conflict of interest statement in the “Confidential to Editor” section, and submit your "Accept" recommendation.

Reviewer #1: (No Response)

Reviewer #2: All comments have been addressed

Reviewer #3: All comments have been addressed

2. Is the manuscript technically sound, and do the data support the conclusions?

Reviewer #1: Partly

Reviewer #2: Yes

Reviewer #3: Yes

3. Has the statistical analysis been performed appropriately and rigorously? 

Reviewer #1: N/A

Reviewer #2: Yes

Reviewer #3: Yes

4. Have the authors made all data underlying the findings in their manuscript fully available?

Reviewer #1: Yes

Reviewer #2: Yes

Reviewer #3: Yes

5. Is the manuscript presented in an intelligible fashion and written in standard English?

Reviewer #1: Yes

Reviewer #2: Yes

Reviewer #3: Yes

6. Review Comments to the Author

Reviewer #1: I appreciate the chance to review the revised manuscript, and commend the authors for their effort in revising it, which have certainly made the paper stronger. In this review, I will address two key areas for further consideration:

1.In my previous comments, I pointed out the need to solidify the theoretical framework, especially regarding the three-factor language model. This area still needs some work, both in the literature review and analysis. The authors themselves have noted conceptual similarities across different constructs and approaches (see line 102). It’s important, then, to clearly demarcate each factor, as this is central to the project. I think that clarifying the boundaries and interactions among these factors would not only strengthen the theoretical underpinnings but also aid in the interpretation of your results. On a related note, about including & testing factors like self-transcendent emotions, the authors mention that their pre-registered analysis doesn’t allow for testing these separately. However, my sense is that the current practices in pre-registration are quite flexible about adding new variables in data analysis, as long as it’s done openly and with good reason. Given self-transcendent emotions are a key part of eudaimonic well-being, ignoring this aspect could be a big miss (especially since it's somewhat hinted in lines 102-105.)

2.Regarding my earlier suggestion to include covariates like “mental health” and “loneliness” as trait factors, the authors note that adding more traits like personality could cloud the relationship between language and well-being indicators. I actually think this is why we should include them. If trait factors account for a substantial variance in well-being, it raises questions about the necessity of examining complex language usage. Conversely, if language markers still account for a significant variance, it substantiates the project's objective. Plus, most tools for psychological language analysis, like LIWC, rely on a simple bag-of-words approach, which doesn’t capture all the nuances of human language. So, adding in these control variables would probably make the model even stronger.

Reviewer #2: Thank you for addressing my concerns. I do not have more comments. This is a very interesting article.

Reviewer #3: Thank you for the opportunity to review the revised manuscript, which proposes a novel conceptual model of language markers of well-being, validated through a cross-sectional study using narratives and LIWC. The work contributes to the field by showing an approach to measuring well-being beyond traditional self-report measures. The methodology, involving the analysis of written narratives through LIWC to validate the conceptual model, is robust. I have a minor comment.

1. While the authors briefly mention the future research needed to validate language markers of well-being, expanding on potential applications of the findings in real-world settings is also valuable.

7. PLOS authors have the option to publish the peer review history of their article (what does this mean?). If published, this will include your full peer review and any attached files.

Reviewer #1: No

Reviewer #2: No

Reviewer #3: No

---

## [Author Response · Author response to Decision Letter 1]

19 Mar 2024

(Please see attached file 'Response to Reviewers')

Dear Dr. Meier,

Thank you for submitting your manuscript to PLOS ONE. After careful consideration, we feel that it has merit but does not fully meet PLOS ONE’s publication criteria as it currently stands. Therefore, we invite you to submit a revised version of the manuscript that addresses the points raised during the review process.

We look forward to receiving your revised manuscript.

Kind regards,

Michal Ptaszynski, PhD

Academic Editor

PLOS ONE

Dear Dr. Ptaszynski,

Thank you for the opportunity to revise our manuscript. We respond to each remaining point raised by the reviewers below, and have made corresponding changes to the manuscript. Additionally, we carefully proofread our manuscript and edited it for clarity and grammar. We hope that the manuscript now meets the publication standards of PLOS ONE.

Thank you very much, again, for considering our manuscript.

Response to Review Comments

(Page numbers refer to the marked-up copy of the manuscript; relevant sections are highlighted in yellow)

Reviewer #1: I appreciate the chance to review the revised manuscript, and commend the authors for their effort in revising it, which have certainly made the paper stronger. In this review, I will address two key areas for further consideration:

1.In my previous comments, I pointed out the need to solidify the theoretical framework, especially regarding the three-factor language model. This area still needs some work, both in the literature review and analysis. The authors themselves have noted conceptual similarities across different constructs and approaches (see line 102). It’s important, then, to clearly demarcate each factor, as this is central to the project. I think that clarifying the boundaries and interactions among these factors would not only strengthen the theoretical underpinnings but also aid in the interpretation of your results.

Response:

We have significantly revised the introduction section (p. 4-8) of our manuscript to solidify our theoretical framework. Thank you for the giving us the opportunity to clarify the aim of a conceptual and empirical integration of the literature of language markers and self-reported well-being. Specifically, we have revised this section to clarify that our three-factor language model is informed by prior evidence that linked word use to affective, evaluative, and social processes of well-being. Additionally, we review the literature on conceptualizations of affective, evaluative, and social well-being, allowing us to conceptually integrate our language markers into the well-being literature. Importantly, we now clarify the prevailing perspective in the literature that language and self-reported well-being represent different entities that provide different types of access to well-being (i.e., with self-report measures explicitly asking about self-perceptions of well-being and language analysis tapping into the less conscious processes that contribute to well-being; Mehl et al., 2017; Boyd & Pennebaker, 2017). While we expect language markers to overlap with self-report scales of well-being, there are reasons to believe that the structure of language marker will somewhat differ from the structure proposed by the well-being literature (which has predominantly relied on self-report scales). Additionally, the current state of the well-being literature reflects the fact that discussions about the specific structure of wellbeing are ongoing (Lambert et al., 2020).

In the revised version, we now clarify that our conceptual language model is informed by prior evidence and conceptualizations of how word use reflects well-being and thus the ways in which well-being and underlying processes are likely to manifest in language use. These concepts are embedded in prominent well-being conceptualizations but not identical to them, as we acknowledge in our conceptual language model. Thank you for raising this point, which helped us to reflect on our theoretical approach and refine it. 

In the analysis (p. 22, 24) and discussion (p. 27-28) parts, we now highlight and reflect on associations between the three latent language factors in more detail. In short, despite some overlap between the language factors (indicated by small-to-moderate associations; Betas between -0.12 and 0.34), each language factor seems to capture unique processes (indicated by the distinct patterns of associations between each language factor and self-reported well-being, as well as non-significant associations between the “affective” and “social” language factor and an overall good model fit) (see p. 21). 

References:

Boyd RL, Pennebaker JW. Language-based personality: a new approach to personality in a digital world. Curr Opin Behav Sci. 2017;18:63–8.

Mehl MR, Raison CL, Pace TWW, Arevalo JMG, Cole SW. Natural language indicators of differential gene regulation in the human immune system. Proc Natl Acad Sci U S A. 2017;114(47):12554–9.

Lambert, L., Lomas, T., van de Weijer, M. P., Passmore, H. A., Joshanloo, M., Harter, J., ... & Diener, E. (2020). Towards a greater global understanding of wellbeing: A proposal for a more inclusive measure. International Journal of Wellbeing, 10(2). https://doi.org/10.5502/ijw.v10i2.1037

On a related note, about including & testing factors like self-transcendent emotions, the authors mention that their pre-registered analysis doesn’t allow for testing these separately. However, my sense is that the current practices in pre-registration are quite flexible about adding new variables in data analysis, as long as it’s done openly and with good reason. Given self-transcendent emotions are a key part of eudaimonic well-being, ignoring this aspect could be a big miss (especially since it's somewhat hinted in lines 102-105.)

Response:

Thank you once again for the valuable suggestion. We indeed agree that exploring self-transcendent emotions, such as gratitude or hope, would be a valuable addition to our research. Unfortunately, it is not possible for us to conduct this additional analysis within the current scope of our manuscript.

Although there is an English version of a self-transcendent emotions LIWC dictionary (Ji & Raney, 2020), all our language samples are in German. Regrettably, there is no existing German dictionary for text analysis on self-transcendent emotions, and developing and validating one is not feasible within the project's scope.

To perform a methodologically sound analysis on our German sample, creating a new German dictionary would be essential. This process involves several rigorous steps, including the collection of additional text samples and data to establish the new dictionary’s psychometric properties (e.g., reliability, equivalence with the original dictionary) [see Marszałek et al., 2023; Meier et al., 2018 for examples of dictionary translations]. A process that we have contributed to for the German version of the established LIWC dictionaries we used in this study (Meier et al., 2018). 

While we acknowledge the importance of self-transcendent emotions in healthy aging, creating a German dictionary and validating its effectiveness is a substantial undertaking that goes well beyond the current study. However, we have significantly expanded our discussion on self-transcendent emotions in the manuscript, elaborating on its potential and relevance in the context of healthy aging and pointing researchers to relevant studies (see p. 32). Thank you for drawing our attention to this important aspect, and we appreciate your understanding of the constraints within our current research scope.

References:

Marszałek, M., Miązek, A., & Roczniewska, M. (2023). Promotion and prevention regulatory focus LIWC dictionary. Polish adaptation and validation. PloS one, 18(7), e0288726.

Meier, T., Boyd, R.L., Pennebaker, J.W., Mehl, M.R., Martin, M., Wolf, M., & Horn, A.B. (2018). “LIWC auf Deutsch”: The Development, Psychometrics, and Introduction of DE-LIWC2015. https://doi.org/10.31234/osf.io/uq8zt

Ji, Q., & Raney, A. A. (2020). Developing and validating the self-transcendent emotion dictionary for text analysis. PloS one, 15(9), e0239050. https://doi.org/10.1371/journal.pone.0239050

2.Regarding my earlier suggestion to include covariates like “mental health” and “loneliness” as trait factors, the authors note that adding more traits like personality could cloud the relationship between language and well-being indicators. I actually think this is why we should include them. If trait factors account for a substantial variance in well-being, it raises questions about the necessity of examining complex language usage. Conversely, if language markers still account for a significant variance, it substantiates the project's objective. Plus, most tools for psychological language analysis, like LIWC, rely on a simple bag-of-words approach, which doesn’t capture all the nuances of human language. So, adding in these control variables would probably make the model even stronger.

Response:

We appreciate your comments and share your recognition of the importance of incorporating trait measures (e.g., loneliness, mental health) in our models. We would like to take this as an opportunity to clarify our approach: 

Validation strategy for language markers. Our strategy went beyond establishing a measurement model of language markers (see above) by validating it through the inclusion of an abundant set of existing, established well-being self-report measures. For this reason, we used 30 established self-report scales (including loneliness and various mental health measures, e.g., depressive symptoms; see Table S5 or below for an overview of all self-report scales included in the combined model) as manifest (observed) indicators of latent well-being factors (psychological well-being, physical health), linking them to language indicators in our combined model (Model 3, p. 24-26 or Fig 2). Thus, instead of including single trait measures as covariates in our models, we used a comprehensive set of 30 self-reported trait measures to (1) create a measurement model and (2) examine the predictive validity of language markers in the combined model. The advantage of using the structural equation modeling (SEM) approach, which allows the estimation of latent factors (and thus a more psychometrically sound estimation of a construct; see MacCallum & Austin, 2000), is that it offers increased robustness (i.e., less prone to measurement error) compared to the inclusion of single covariates.

This is where we see our study’s main contribution: Our study aims to validate language markers by linking them with self-reported items of established scales. In doing so, our focus is on contributing to the understanding of LIWC indicators, which have been used very frequently in research, despite their limitations as a bag of word-approach. Thus, rather than maximizing overall fit, we prioritized the validation of these theoretically informed and commonly used language markers.

Language analysis offers opportunities for at-scale analysis of behavioral data that do not come with the same issues as traditionally-used self-report measures. Our findings demonstrate that language markers align with self-reported trait measures but exhibit unique features, showcasing the added value of language markers (for a review see Boyd & Pennebaker, 2017). These findings are relevant in the context of larger trends in natural language processing such as LLM (large language models). 

Acknowledging the need for clarification, we have made several revisions to the manuscript (e.g., p. 13, 17-18) to better explain our approach. We now report R squared to address the reviewer’s concern about variance accounted for by language indicators (vs. self-report/trait measures), and critically discuss them (p. 25, 34). 

In summary, our approach aligns with the reviewer’s emphasis on connecting language measures with established trait measures (e.g., loneliness, mental health). Our models incorporate a comprehensive set of 30 self-report scales, account for measurement error (i.e., by combining several self-report scales in a measurement model with latent factors), and present a structure of latent concepts based on language and self-report indicators. Thank you for providing an opportunity to clarify our approach, and we hope these revisions address your concerns.

References

Boyd, R. L., & Pennebaker, J. W. (2017). Language-based personality: A new approach to personality in a digital world. Current opinion in behavioral sciences, 18, 63-68. https://doi.org/10.1016/j.cobeha.2017.07.017

MacCallum, R. C., & Austin, J. T. (2000). Applications of structural equation modeling in psychological research. Annual Review of Psychology, 51, 201–226. https://doi.org/10.1146/annurev.psych.51.1.201

Overview of all 30 self-report scales included in the combined model (see Table S5):

• Psychological Health (WHO QoLBref)

• Life Satisfaction (current)

• Depression (PHQ9)

• Life Satisfaction (SWLS)

• Self-Acceptance (Ryff scale)

• Life Satisfaction (retrospective)

• Life Satisfaction (prospective)

• Mental Health (MHI4P; SF-12)

• Quality of Life (FQOL)

• Environment (WHO QoLBref)

• Vitality (SF-12)

• Environmental Mastery (Ryff Scale)

• Mental Health (MHI3; SF-12)

• Positive Affect (PANAS)

• Quality of Life (SEIQoL)

• Social Relationships (WHO QoLBref)

• Social Functioning (SF-12)

• Loneliness (UCLA)

• Positive Relationships (Ryff Scale)

• Perceived Social Support (Instrumental; BSSSI)

• Perceived Social Support (Emotional; BSSSE)

• Personal Growth (Ryff Scale)

• Negative Affect (PANAS)

• Global Quality of Life (WHO QoLBref)

• Physical Health (WHO QoLBref)

• Physical Health (GHP1; SF-12)

• Sleep Quality (Jenkin's scale)

• Pain (SF-12)

• Physical Functioning (PFI02P; SF-12)

• Physical Functioning (PFI04P; SF-12)

Reviewer #2: Thank you for addressing my concerns. I do not have more comments. This is a very interesting article.

Response: Thank you!

Reviewer #3: Thank you for the opportunity to review the revised manuscript, which proposes a novel conceptual model of language markers of well-being, validated through a cross-sectional study using narratives and LIWC. The work contributes to the field by showing an approach to measuring well-being beyond traditional self-report measures. The methodology, involving the analysis of written narratives through LIWC to validate the conceptual model, is robust. I have a minor comment.

1. While the authors briefly mention the future research needed to validate language markers of well-being, expanding on potential applications of the findings in real-world settings is also valuable.

Response: We have now added examples of a possible real-world application as a future direction (p. 34). Thank you for suggesting this.

---

## [Decision Letter · Decision Letter 2]

27 Mar 2024

When I am Sixty-Four… Evaluating language markers of well-being in healthy aging narratives

PONE-D-23-15452R2

Dear Dr. Meier,

We’re pleased to inform you that your manuscript has been judged scientifically suitable for publication and will be formally accepted for publication once it meets all outstanding technical requirements.

Kind regards,

Michal Ptaszynski, PhD

Academic Editor

PLOS ONE

Additional Editor Comments (optional):

Reviewers' comments:

Reviewer's Responses to Questions

**Comments to the Author**

1. If the authors have adequately addressed your comments raised in a previous round of review and you feel that this manuscript is now acceptable for publication, you may indicate that here to bypass the “Comments to the Author” section, enter your conflict of interest statement in the “Confidential to Editor” section, and submit your "Accept" recommendation.

Reviewer #1: All comments have been addressed

2. Is the manuscript technically sound, and do the data support the conclusions?

Reviewer #1: Yes

3. Has the statistical analysis been performed appropriately and rigorously? 

Reviewer #1: Yes

4. Have the authors made all data underlying the findings in their manuscript fully available?

Reviewer #1: Yes

5. Is the manuscript presented in an intelligible fashion and written in standard English?

Reviewer #1: Yes

6. Review Comments to the Author

Reviewer #1: I commend the authors for the efforts demonstrated in their revisions. At this point, I am confient to recommend your manuscript for publication at PLOS ONE. With just a minor lanauge note:

line, 386: The fit indices recommended by [56] were used to evaluate model fit... given the citation style, this may best be put along the lines of: "The fit indices used to evaluate model fit include ..."

7. PLOS authors have the option to publish the peer review history of their article (what does this mean?). If published, this will include your full peer review and any attached files.

Reviewer #1: No

---

## [Editor Report · Acceptance letter]

3 Apr 2024

PONE-D-23-15452R2 

PLOS ONE

Dear Dr. Meier, 

I'm pleased to inform you that your manuscript has been deemed suitable for publication in PLOS ONE. Congratulations! Your manuscript is now being handed over to our production team.

Kind regards, 

on behalf of

Dr. Michal Ptaszynski 

Academic Editor

PLOS ONE